# Comparative Effects of Milk Containing A1 versus A2 β-Casein on Health, Growth and β-Casomorphin-7 Level in Plasma of Neonatal Dairy Calves

**DOI:** 10.3390/ani11010055

**Published:** 2020-12-30

**Authors:** Lisa G. Hohmann, Tong Yin, Helen Schweizer, Isabella J. Giambra, Sven König, Armin M. Scholz

**Affiliations:** 1Institute of Animal Breeding and Genetics, Justus-Liebig-University of Gießen, Ludwigstr. 21b, 35390 Giessen, Germany; tong.yin@agrar.uni-giessen.de (T.Y.); isabella.j.giambra@agrar.uni-giessen.de (I.J.G.); sven.koenig@agrar.uni-giessen.de (S.K.); 2Livestock Center Oberschleissheim, Veterinary Faculty of the Ludwig-Maximilians-University Munich, St. Hubertus Str. 12, 85764 Oberschleissheim, Germany; Helen.Schweizer@lvg.vetmed.uni-muenchen.de (H.S.); armin.scholz@lvg.vetmed.uni-muenchen.de (A.M.S.)

**Keywords:** A2-milk, beta-casomorphin-7, calf health, caseins, growth, milk protein polymorphism

## Abstract

**Simple Summary:**

Bovine milk generally contains two types of the milk protein β-casein, A1 and A2. Enzymatic digestion of the A1 type yields the opioid peptide β-casomorphin-7, which is suggested to adversely affect human and animal health. This study aimed to compare the effects of milk containing either homozygote A1 or A2 β-casein on health and growth parameters in 47 dairy calves during the first three weeks of life. Additionally, we studied, for the first time, the levels of intact β-casomorphin-7 in plasma of calves fed milk of alternative β-casein genotypes. Milk feeding of “A2-milk” led to a lower milk intake and a looser fecal consistency (higher prevalence of diarrhea) compared to “A1-milk”. Nevertheless, weight gains and end weights of calves of both feeding groups were similar, which might be caused by the associated higher protein content of milk containing the A2 variant. Intact β-casomorphin-7 was detected in plasma after A1- and A2-milk consumption, but was almost 5 times higher for A1-calves. In summary, A2-milk minimized the cleavage of the opioid peptide β-casomorphin-7 and might have advantages in the development of pre-weaned dairy calves.

**Abstract:**

Research has shown that digestion of A1 β-casein (β-CN) affects gastrointestinal motility and opioid activity through the release of the peptide β-casomorphin-7 (β-CM7). In the case of the A2 variant, the cleavage of β-CM7 does not occur or occurs at a very low rate. Therefore, the aim of the study was to compare the effects of milk containing either homozygote A1 or A2 β-CN on health and growth parameters of dairy calves. Forty-seven neonatal calves (24 females, 23 males) of the breeds German Holstein (GH, *n* = 9), German Simmental (GS, *n* = 33) and their crossing (GH × GS, *n* = 5) were used in a 21-day feeding study. Fecal score (FS), respiratory frequency (RF), and rectal body temperature (BT) were recorded daily, whereas body weight was measured at birth and at day 21 to estimate the average daily weight gain (ADG). Additionally, blood was collected from calves three times during the experimental period and, for the first time, the respective plasma samples were analyzed for intact β-CM7. Consumption of A2-milk led to a lower daily milk intake (dMI) (*p* < 0.05). Furthermore, fecal consistency was softer for calves fed A2-milk (*p* < 0.05). Although 44% of A2-calves had diarrhea or revealed a tendency towards it (FS ≥ 3), A1-calves had a prevalence of 21%. Calves with a FS of 4 were offered an electrolyte solution and received a dietary food supplement for the stabilization of the fluid and electrolyte balance. Nevertheless, similar ADG and end weights (EW) of calves fed A1- or A2-milk (*p* > 0.05) indicate that A2-milk may compensate higher diarrhea rates and lower dMI due to the associated higher protein content. This is the first report of intact β-CM7 in plasma of calves fed milk of either A1 or A2 β-CN. Evidence from this study suggests that due to the change in the amino-acid sequence, A2-milk might be able to prevent or, at least, to minimize the cleavage of β-CM7 in calves.

## 1. Introduction

The pre-weaning period is critical for calf health and growth. Especially neonatal calf diarrhea is the primary mortality cause, resulting in substantial economic losses in the dairy industry [1]. Milk feeding programs considering milk composition may assist postnatal development through intensive nutrient intake and elevated energy level to improve structural growth, health, and vitality [2,3]. Proteins contained in milk—especially the four caseins α_s1_-casein (α_s1_-CN), β-casein (β-CN), α_s2_-casein (α_s2_-CN) and κ-casein (κ-CN)—exhibit immunomodulatory [4,5], antihypertensive [6], antimicrobial [7], antioxidative [8] and opioid-like peptides [9]. At present, there is an ongoing debate about the health-impairing effects of the A1 allele of β-CN in milk compared with the progenitor A2 allele. These variants differ at a single amino-acid position 67, being a Histidine in A1 and a Proline in A2 casein [10]. Because of this substitution, enzymatic digestion of the variant A1 yields the opioid peptide β-casomorphin-7 (β-CM7) [11]. In the case of A2 β-CN, the enzymatic hydrolysis of the bond between Ile^66^ and Pro^67^ does not occur or occurs at a very low rate [12].

Animal data clearly indicate that β-casomorphins, including β-CM7, can act as opioid receptor agonists, probably acting via μ-type opioid receptors. First feeding experiments in this regard were conducted in Australia, indicating that A1 β-CN was atherogenic (in comparison with A2) [13]. Haq et al. and Barnett et al. concluded that the consumption of A1 variants of β-CN induces inflammatory response in mice and rats’ gut by activating the Th2 pathway [14,15]. Hedner and Hedner showed that β-CM7 and its derivative (β-CM5) induce apnea and irregular breathing in adult rats and newborn rabbits [16]. Furthermore, β-CM7 induces a variety of effects on gastrointestinal functions, e.g., effecting the frequency reduction and amplitude of intestinal contractions [17,18] and mucus secretion [19]. Thus, β-CM7 increases the total gastrointestinal transit time, and accordingly, prevents diarrhea [20,21]. In addition, a recent human study [22] suggested significantly higher stool consistency values when consuming milk containing A1 compared to A2 β-CN. However, all information available is based on either in vitro trials or in vivo data taken from rats, mice, rabbits and humans [23,24]. In most if not all animal studies to date, β-CM7 concentrations of treatments exceeded native concentrations of bovine milk, or β-CM7 was administrated intra-peritoneal or intra-cerebroventricular.

For opioid activity in the central nervous system after oral ingestion, the passage of A1-milk derived β-CM7 through the intestinal mucosa, and moreover through the blood-brain-barrier, is needed [25]. However, the mechanisms of the transfer of intact peptides longer than tripeptides across the intestinal barrier are mostly unclear. The presence of β-CM7 immunoreactive material (β-CM7_IRM_) has been reported in blood in two studies with neonatal calves and dogs [26,27]. Umbach et al. found no β-CM7_IRM_ in samples collected before first milk intake. In contrast, in samples collected after milk intake, β-CM7_IRM_ was detected [26]. Singh et al. found that β-CM7_IRM_ increased significantly in both 2- and 4-weeks-old puppies post bovine milk feeding, while β-CM7_IRM_ levels were undetectable in adult dogs before or after bovine milk feeding [27]. Both studies indicated that the detected β-CM7_IRM_ consists of 12–13 amino-acid residues and might represent a precursor of the heptapeptide β-CM7 [26,27]. However, the presence of intact β-CM7 molecules in blood after intake of milk has not been proven in in vivo studies. In certain cases, such as in neonates, the intestinal mucosa is more permeable to relatively large peptides during postnatal formation, making them more likely candidates than adults to experience the opioid effects of β-CM7 from bovine casein [28].

To our knowledge, no single study has investigated effects of A1-milk and its native concentrations of digested β-CM7 on health parameters in neonatal calves. We hypothesize that the consumption of A1-milk leads to increased β-CM7 plasma levels with adverse effects on the development of neonatal calves. Great relevance to practical agriculture is given, as milk is the main source of both nutritive and biologically active material for newborn calves. Therefore, the objective of the present study was to evaluate comparative effects of feeding either homozygote A1- or A2-milk on growth and health parameters of neonatal dairy calves. The study also presents the first comprehensive information on intact β-CM7 in plasma after milk intake of different β-CN types.

## 2. Materials and Methods

The experiment was conducted at the Livestock Center Oberschleissheim, Veterinary Faculty of the Ludwig-Maximilians-University of Munich and was performed from July 2019 to February 2020. The housing and treatment of the animals were carried out in accordance with the national and international animal welfare laws. All methods presented in the study were non-invasive and belong to routine practices. Therefore, they did not cause pain, suffering, or harm to the animals of this study in compliance with the German Animal Welfare Act §7. Nevertheless, the presented procedures have been approved for a subsample of calves that were used for additional blood parameter analyses by the District Government of Upper Bavaria (Az. 3532.Vet_03-19-31).

### 2.1. Milk Protein Typing

Milk samples from 111 dairy cows of the breeds German Holstein (GH), German Simmental (GS), and their crossbred product (GH × GS) were analyzed for milk protein polymorphisms of the caseins α_s1_-CN, β-CN, α_s2_-CN and κ-CN by isoelectric focusing on ultrathin polyacrylamide gels using carrier ampholytes according to Seibert et al. and Erhardt [29,30].

Cows differing only in the β-CN locus (homozygote A1A1 versus homozygote A2A2 genotypes) were selected for separated milking via a Lely robotic milking system M4USE (Lely Deutschland, Waldstetten, Germany). The milk of these cows built the basis for the milk feedings of calves under study.

### 2.2. Animals, Housing, and Feeding

Forty-seven neonatal calves (24 females, 23 males) of the breed groups GH, GS, and GH × GS were randomly assigned to one of two feeding groups (FG): (1) an A1-milk diet with homozygote β-CN genotype A1A1 (*n* = 26), or (2) an A2-milk diet with homozygote β-CN genotype A2A2 (*n* = 21). Calves from both groups were used to evaluate several health and growth traits during the first three weeks of life. After birth, the calves were removed from their mother and received the usual postnatal care, as drying of the newborns and navel care. Maternal colostrum was fed within 2 h postpartum (first milking) and for the following two feedings from open, 8 L plastic buckets. Starting with the fourth feeding, all calves received milk of either A1A1 or A2A2 genotype. Feeding times were 3 times daily; with 3 L at 06.00, 1 L at 11.30, and 3 L at 17.30. At each feeding per day, the actual milk intake (liters of milk offered minus leftover) was recorded and summed up for the daily milk intake (dMI) per calf. Calves were housed in pairs in weatherproof calf igloos bedded on straw. Calves had *ad libitum* access to water and hay but had no access to calf starter. As the hay intake of all calves during their first three weeks of life is negligibly small, it does not have to be considered in the feeding ratio. At an age of 16 days, the calves were dehorned using sedation with local anesthetic. For pain relief after dehorning, calves received a single injection of 0.4 mg meloxicam per kg body weight.

### 2.3. Experimental Measurements

During the experimental period of 21 days, calves were weighed at birth and at the last day of the study (day 21). Average daily weight gain (ADG) from birth weight (BW) to end weight (EW) was calculated as (EW-BW)/(21 days). The following health parameters were monitored daily: Rectal body temperature (BT) was recorded for each calf—with temperatures > 39.5 °C considered elevated. Fecal consistency, using fecal score (FS), was assessed according to Wickramasinghe et al. [31]: 1 = normal, 2 = soft, 3 = runny, or 4 = watery/diarrhea. Calves with a FS of 4 were offered an electrolyte solution and received an extra treatment of a paste containing bentonite-montmorillonite and minerals (Enterogelan^®^, Virbac Tiergesundheit, Bad Oldesloe, Germany). Respiratory parameters included respiratory disease, respiratory frequency (RF) and nasal discharge. The scoring varies between 0 = hot, dry nose, 1 = normal, 2 = runny nose and 3 = heavy breathing/cough, whereas the RF was measured by breaths per minute (bpm). All parameters were measured by the same person to maintain consistency of measurements each day. Data regarding calving and veterinary therapy (i.e., birth type, veterinary obstetrics and treatment) were documented by the veterinarian.

As physiological variables including RF and BT might be affected by meteorological effects, the mean ambient temperature (Ta), mean relative humidity (RH) and sunshine hours (sun) per day were accurately transferred from German Meteorological Services. The daily temperature-humidity index (THI) was calculated using the following equation by Kibler [32]: THI = 1.8Ta − (1 − RH) (Ta − 14.3) + 32. The THI values were categorized according to the thermal neutral zone for neonatal calves as follows: cold stress, THI1 < 35; thermoneutral, THI2 ≥ 35 and < 65; heat stress, THI3 ≥ 65.

### 2.4. Blood Sampling

Blood was collected from 34 calves three times during the experimental period. The first blood sampling (B1, *n* = 15) was performed right after calves’ birth just before first colostrum intake. Calves that were born during the night and provided with colostrum before blood was collected were excluded from the sampling at B1. The second (B2, *n* = 33) and third blood samples (B3, *n* = 34) were collected 2 to 3 h after morning feeding on days three and 21 of life, respectively. Blood was drawn from the jugular vein into EDTA-coated tubes (Monovette^®^, Sarstedt, Numbrecht, Germany). Blood was immediately centrifuged at 2000× *g* at 4 °C for 25 min. Blood plasma was separated and stored in a −20 °C freezer until subsequent analyses.

### 2.5. Analysis of β-Casomorphin-7 from Plasma

Plasma levels of the peptide β-CM7 were measured using a commercial ELISA-Kit (Creative Diagnostics, New York, NY, USA) and a Bio-Rad *iMark 1.04.02* microplate reader (Bio-Rad, Neuberg, Germany), in accordance with the manufacturer’s instructions. The calculated intra-assay variability was 5.7%, and the inter-assay variability determined by testing the same set of samples on a different plate was 11.1%.

### 2.6. Phenotypic Data of the Calves

Means and standard deviations of BW, ADG, EW, dMI, BT, and RF of calves under study are presented in Table 1. GH calves revealed the heaviest BW (44.3 kg), followed by GS (43.7 kg) and crossbred calves (41.1 kg). Daily milk intake was highest for GH calves (6.78 L) resulting in highest ADG with 0.71 kg/day. GS calves consumed the lowest amount (6.53 L) of milk per day. The RF was highest for GH (52 bpm) and lowest for GS (44 bpm), whereas crossbred calves behaved intermediately with 44 bpm. Rectal BT of calves were similar for all breed groups (38.8 °C).

### 2.7. Statistical Analyses

#### 2.7.1. Impact of Milk Feeding Group (A1- vs. A2-Milk) on Milk Intake and Health Status of Calves

Effects of the FG on RF, BT, FS, and dMI were analyzed applying the following linear mixed model:(1)$$y_{\mathrm{ijkmn}}\;=\;Breed\;{group}_{i}\;+{Sex}_{j}\;+\;{THIclass}_{k}\;*Sun\;+\;dMI\;+\;\sum_{l=0}^{2}\alpha_{ml}FG_{m}\;+\;C_{n}\;+\;e_{\mathrm{ijkmn}}$$
where *y*_ijkmn_ = observations for RF, BT, FS, and dMI: *Breed group*_i_ = fixed effect of the breed group (GH, GS, GH × GS); *Sex*_j_ = fixed effect of sex (female, male); *THIclass*_k_ = fixed effect of THI class (THI1, THI2, THI3); *Sun* = linear regression on sunshine hours per day; *dMI* = linear regression on dMI (apart from the model where dMI was the trait of interest); *FG*_*m*_ = fixed effect of FG (A1, A2); *a*_*ml*_ = fixed regression coefficients using second order Legendre polynomials at day 1 to 21 nested within FG; *C*_n_ = random effect of the calf considering repeated measurements; and *e*_ijkmn_ = random residual effect. Interactions among THI classes and sunshine hours per day were initially tested, but were not included for the trait dMI as they were not statistically significant (*p* > 0.05).

#### 2.7.2. Impact of Milk Feeding Group (A1- vs. A2-Milk) on Growth Traits of Calves

For the traits ADG and EW, the statistical model (2) was defined as follows:*y*_ijkl_ = *Breed group*_i_ + *Sex*_j_ + *THI* + *tMI* + *BW* + *type*_k_ + *FG*_l_ + *e*_ijkl_(2)
where *y*_ijkl_ = observations for ADG and EW: *Breed group*_i_ = fixed effect of breed group (GH, GS, GH × GS); *Sex*_j_ = fixed effect of sex (female, male); *THI* = linear regression for the average THI during the experimental period of 21 days; *tMI* = linear regression for the total milk intake (tMI) during the experimental period of 21 days; *BW* = linear regression on the BW of calves; *type*_k_ = fixed effect of birth type (single or twin); *FG*_l_ = fixed effect of FG (A1, A2); and *e*_ijkl_ = random residual effect.

Statistical analyses were conducted using the software R 2.14.2 [33], applying the packages lme4 [34] and lmerTest [35] for fitting linear mixed-effects models. The package emmeans [36] was used for the calculation of least squares means. Significant differences were declared at *p* < 0.05.

#### 2.7.3. Quantification of β-Casomorphin-7 in Plasma

The mean and standard error of the β-CM7 level in plasma of calves per feeding group (A1, A2) for each blood sampling (B1, B2, B3) were calculated. Statistical significance of means between both feeding groups for each blood sampling was assessed by a Student’s *t*-test implemented in the R package stats [33]. *p*-values ≤ 0.05 were considered significant.

## 3. Results and Discussion

### 3.1. Allele and Genotype Frequencies at the β-Casein Locus

There was a predominance of the A1 (46%) and A2 (51%) alleles as shown by the presence of A1A1, A1A2 and A2A2 β-CN genotypes in about 95% of all milk samples (Table 2). The rare variants A3 and C were not detected in the cows studied. Only the variant B occurred with an average low frequency of 3% across the breeds. In comparison to the breed GH, GS cattle revealed a higher frequency for the favored A2 variant. This agrees with results by Meier et al. [37].

Cows differing only in the β-CN locus carry most frequently either the composite casein genotype BBǀA1A1ǀAAǀAA (order of genes on bovine chromosome 6: α_s1_-ǀβ-ǀα_s2_-ǀκ-CN) or BBǀA2A2ǀAAǀAA. Frequencies of 8.8% and 4.9% were observed for the composite genotypes BBǀA1A1ǀAAǀAA and BBǀA2A2ǀAAǀAA, respectively.

### 3.2. Impact of the Milk Feeding Group (A1- vs. A2-Milk) on Milk Intake and Growth Traits of Calves

#### 3.2.1. Daily Milk Intake

Milk feeding effects on dMI (model 1) are given in Table 3. Calves fed A1-milk consumed a significantly greater amount of milk (7.28 L/day; *p* < 0.05) than calves receiving A2-milk (6.96 L/day). Summed up for the first three weeks of life, the milk intake of the respective milk feeding groups notably differed by 6.51 L (151.83 L for A1-calves; 145.32 L for A2-calves). One hypothesis for the lower milk intake of A2-calves is that milk containing β-CN A2 includes a higher protein content. Several studies addressing associations between milk protein variants with milk production traits detected a higher protein content and protein percentage of milk containing the homozygote A2 genotype of β-CN compared to milk containing other β-CN variants [38]. In agreement with present results, Morrison et al. reported higher intakes of milk replacer containing a lower crude protein content compared to milk replacer containing a higher crude protein content during the pre-weaning phase of Holstein Friesian calves [39].

Figure 1 shows temporal changes of dMI over the first three weeks of life. Daily milk intake of both FG increased over two weeks. Between days 15 and 18, the calves were anesthetized and dehorned explaining their decreasing intake of milk. Compared to A1-calves, calves receiving A2-milk revealed a slighter reduction in milk consumption after dehorning and stabilized quicker.

#### 3.2.2. Growth Performance

Results from model 2 revealed that the FG does not have a significant impact on ADG and on EW of calves (Table 3; *p* > 0.05). Calves fed with A1-milk gained 0.75 kg/day, whereas calves fed A2-milk gained 0.64 kg/day (Figure 2a). Additionally, least squares means for EW of A1- and A2-calves were similar (Table 3; *p* > 0.05). At an age of 21 days, calves fed A1- and A2-milk weighed 59.4 kg and 57.0 kg, respectively (Figure 2b).

A1-calves consumed a significantly greater amount of milk than A2-calves. However, A2-calves may have abilities to compensate their lower milk intake to achieving a similar growth performance (ADG, EW) as observed for A1-calves. One explanation in this regard is that A2-milk features a higher protein content and provides the protein requirements of rapidly growing muscle [40]. On the other hand, the milk conversion ratio (milk intake (kg)/weight gain (kg)) during the pre-weaning period might be different between calves receiving A1- and A2-milk. The results revealed that the milk conversion ratio was larger for A2-calves compared to A1-calves (10.5 and 9.2, respectively). In this context, Sharma et al. recently reported that feed conversion efficiency was significantly improved with higher protein level [41]. Additionally, the authors indicated that the nutrient digestibility increased with an increasing protein level.

The effects sex, birth type, BW, and tMI significantly influenced EW of calves. Female calves tended to weigh 4.5 kg less than male calves at day 21 (*p* < 0.1). Several studies revealed lower body weights of female calves compared to male calves due to distinct sexual dimorphism in cattle [42,43]. Calves born as twins revealed lower EW (45.4 kg) compared to single born calves (58.1 kg; *p* < 0.05), because twins generally have a lower BW due to the placental development as indicated by previous studies [42,43].

### 3.3. Impact of the Milk Feeding Group (A1- vs. A2-Milk) on Health Parameters of Calves

#### 3.3.1. Rectal Body Temperature

The results from model 1 indicate that BT did not differ significantly between both FG (*p* > 0.05). The control of a quite constant BT reflects the thermoregulatory mechanism of homeothermic animals [44]. Nevertheless, if the ambient temperature exceeds the thermoneutral zones of calves (13–25 °C) along with high humidity and slow air movement, calves cannot dissipate the excess of heat adequately and BT increases [45]. Beakley and Findlay reported increasing BT of animals with increasing environmental temperature and humidity [46]. In the present study, the interaction among THI classes and sunshine hours per day showed a significant effect on BT (*p* < 0.05). However, least squares means of BT for the THI classes were similar (THI1 = 38.9 °C, THI2 = 39.0 °C, THI3 = 38.9 °C; *p* > 0.05). Similar BT responses on THI were expected, because the THI did not exceed a value of 74 during the experimental period between July and February. Kovács et al. inferred that welfare of young calves is compromised above a THI of 78, and that calves experience significant heat stress above a THI of 88 [47]. Therefore, the calves were not exposed to extremely elevated ambient temperatures and humidity in the present study.

#### 3.3.2. Respiratory Frequency

The FG did not affect RF of calves significantly (*p* > 0.05; Table 3). The average RF of 47 and 53 bpm detected for A1- and A2-calves, respectively, represent the standard values for bovine neonates [48,49]. Nevertheless, a difference between both FG was observed for RF as a function of time (age in days) (Figure 3). Although the RF of A2-calves increased over 9 days and afterwards decreased slightly, A1-calves showed a linear RF decline during the experimental period. It is interesting to note that bovine β-casomorphins were associated with apnea and irregular breathing [20,50]. The opioid effects, however, have only been observed following intra-cerebroventricular administration. For opioid activity in the central nervous system after oral ingestion, the passage of A1-milk derived β-CM7 through the blood-brain-barrier is in principle needed. Overall, temporal changes in RF of both FG can be explained by the incomplete development of the anatomical and functional roles of newborns. Linke et al. showed that healthy bovine neonates need at least two weeks before all lung units are integrated into the gas exchange [51]. A similar study with Limousin calves also reported changes in the RF during the first month of life, displaying the extra-uterine physiological adaptation during the neonatal period [49].

Furthermore, the results revealed that the meteorological effects (THI and sun) as well as BT had a significant impact on RF. The explanation addresses physiological processes of thermoregulation [52]. Heat loss (i.e., by respiration) is a response to increasing ambient temperature and simultaneously an effect of homeothermy. Physical responses to heat stress in dairy cows are breed specific [44]. In consequence, we expected a difference in RF between the breed groups. Interestingly, GS tended to have a lower RF compared to GH (49 and 56 bpm, respectively; *p* < 0.1). The reason might be an improved heat tolerance of Simmental cattle. Gantner et al. confirmed that GS cows were less sensitive to thermal stress than Holstein Friesian [53].

#### 3.3.3. Fecal Consistency

Figure 4 reveals that the FS peaked during the second week of life in both FG, but slightly declined afterwards until the end of the experimental period. Generally, calves are susceptible to neonatal calf diarrhea especially during the first 28 days of their life, whereas the most crucial age for the occurrence of neonatal calf diarrhea is the sixth day of life [54]. As indicated in Table 3, the FG significantly influenced fecal consistency of the calves. Calves fed with A2-milk had a lower fecal consistency compared to calves fed with A1-milk (*p* < 0.05, Table 3).

Figure 5 shows the prevalence of fecal scores of calves according to their FG. The prevalence of A1- and A2-calves suffering from diarrhea (FS = 4) was 6 and 10%, respectively. Additionally, the prevalence of FS of 3, which is an indicator towards diarrhea, was more than two times higher in A2-calves compared to A1-calves (34 and 15%, respectively).

A recent human study also detected significantly higher stool consistency values after consumption of milk containing the bovine β-CN variant A1 compared with the variant A2 [22]. In this regard, several studies reported impact of A1-milk derived β-CM7 on gastrointestinal functions via opioid-dependent pathways [18]. β-casomorphins have been demonstrated to prolong gastrointestinal transit time [18,55] and to increase mucus secretion [19]. Furthermore, β-casomorphins enhance net water and electrolyte absorption in the small and large intestine, which is also a major component of their anti-diarrheal action [18,56].

### 3.4. β-Casomorphin-7 Level in Plasma of Calves after A1- or A2-Milk Intake

Blood samples were collected from neonatal calves immediately after birth and on days three and 21 of life, and the respective plasma samples were analyzed for the presence of intact β-CM7. In 13 of the 15 samples collected after birth (B1), no β-CM7 was detected, indicating no difference in β-CM7 level between the FG (*p* > 0.05, Figure 6). This result was expected, as β-CM7 is a milk opioid peptide derived from bovine β-CN, but calves received their first colostrum feeding only after blood sampling B1. The mean β-CM7 level of 1.35 ng/mL in the remaining two samples is because these calves were born during the night, so that the intake of colostrum was ensured before blood (B1) was collected.

However, intact β-CM7 was found in all plasma samples (*n* = 33) obtained after three days of life (B2). This implies the stability against peptidases and the passage of the seven amino-acid-long peptide β-CM7 across the intestinal barrier into the blood circulation. Relatively little is known on the mechanisms of transfer of intact peptides longer than three amino acids through the intestinal mucosa. In specific situations (i.e., stress, diseases or aggression), both animals and humans display increased intestinal permeability [25]. In particular, the gut of neonates is almost permeable to proteins explaining the high level of β-CM7 at B2 (Figure 6). Furthermore, neonates feature a sub-maximal activity of peptidases. Thus, the resorption of peptide fragments longer than tripeptides is suspected [57]. Depending on the FG, calves revealed similar levels of β-CM7 at B2 (A1, 66.84 ng/mL; A2, 56.60 ng/mL). Calves were fed for the first three feedings with maternal colostrum, which was not either homozygote A1- or A2-milk justifying the similar observations of β-CM7 level in both FG at B2.

At day 21 (B3), the β-CM7 levels of calves from both FG were lower compared to the β-CM7 levels at B2, which might be due to the decreasing permeability of the intestine to proteins and macromolecules with increasing age. Comparing both FG at B3, the β-CM7 level of calves fed A1-milk significantly exceeds the β-CM7 level in plasma of calves fed A2-milk (*p* < 0.05). Figure 6 demonstrates that the β-CM7 level after A1-milk feeding is almost five times higher in relation to A2-milk feeding (55.82 and 12.73 ng/mL, respectively). The difference arises from 18 days of exclusively either homozygote A1- or A2-milk feeding. Several studies revealed that β-CM7 originates exclusively from β-CN genetic variants A1 or B [58,59], but recent research identified β-CM7 also from milk produced by homozygote A2 cows [12,60]. Nevertheless, Cieślińska et al. revealed that the content of β-CM7 was four times higher in hydrolyzed milk from homozygote A1-cows than in milk produced by A2-cows [12]. This is reflected by the β-CM7 plasma level of the respective FG at B3. Additionally, Nguyen et al. described the natural occurrence of endogenous β-CM7 in raw milk from different breeds [61]. Therefore, a low β-CM7 plasma level is expected after milk consumption, independent from its genetic variants.

### 3.5. Use of β-Casein Polymorphisms in Selection

With globalization of the dairy industry and market and decreasing milk prices, it is of increasing importance to reduce production costs. Lowering age at first calving might be an efficient strategy for the dairy farmer to reduce costs. Schäff et al. indicated that accelerated growth in pre-weaned calves affect the life-time performance of subsequent dairy cows [2]. Therefore, there is great interest in understanding the consequences of intensive milk feeding on the pre-weaning growth, development, and health of calves.

The present results show that calves receiving A2-milk need less milk for similar ADG and EW due to a higher protein content of A2-milk. Thus, farmers are able to sell more milk or moreover, A2-calves can reveal similar EW at a younger age compared to A1-calves. In a long term perspective, effects imply a reduction of age at first calving and a subsequent higher milk production [62].

Furthermore, results of the analysis for β-CM7 in plasma of calves reveal that the level of β-CM7 is almost 5 times higher after 21 days A1-milk feeding compared to A2-milk feeding. Evidence from this study suggests that A2-milk in animals and humans prevents or minimizes the cleavage of β-CM7 (due to its change in amino-acid sequences).

Therefore, selection for the A2 variant of β-CN may offer new perspectives for breeding strategies in dairy cattle to improve pre-weaning calf performance, and moreover for production of milk with special protein content in niche markets (i.e., functional food, infant formulas).

## 4. Conclusions

Our study shows that the FG does not have an impact on the health indicators RF and BT. We assume that the stability of β-CM7 in plasma is strongly limited and thereby the transfer across the blood-brain barrier does not occur for opioid activity in the central nervous system. Nevertheless, we demonstrated the passage of the seven amino-acid-long peptide β-CM7 through the intestinal mucosa, as intact β-CM7 was detected in plasma of calves for the first time. Calves fed A1-milk revealed an almost 5 times higher β-CM7 plasma level compared to calves fed A2-milk. Furthermore, the results reveal that calves receiving A2-milk had a softer fecal consistency and therefore a higher prevalence of diarrhea compared to calves fed A1-milk. Nevertheless, A2-calves obtained similar ADG and EW compared to calves fed A1-milk despite a significant lower milk intake. One explanation may be that A2-calves can compensate this loss with a superior balance of amino acids, which is constituted by the associated higher protein content in milk containing the A2 variant. Further research is needed for more conclusive evidence. If no antagonistic effects occur, selection for the A2 variant of β-CN may generate high quality milk to improve pre-weaning calf performance. These findings will also have an impact on human health by contributing new prospects to the field of functional food, infant formulas, or other pharmaceutical products.

## Figures and Tables

**Figure 1 animals-11-00055-f001:**
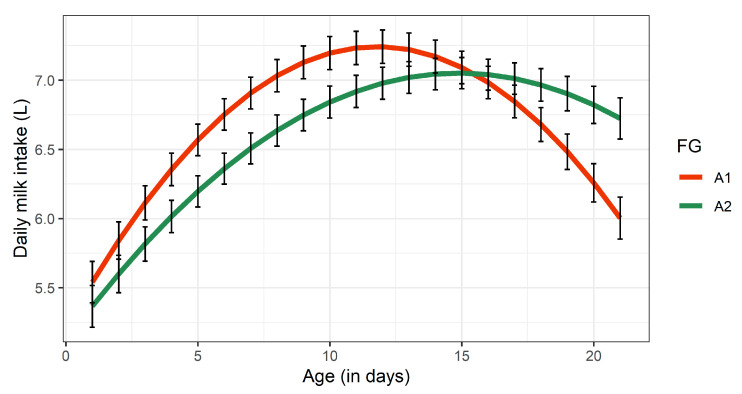
Least squares means for daily milk intake of calves receiving either A1- or A2-milk during the experimental period of 21 days. Black bars indicate standard errors. FG, feeding group.

**Figure 2 animals-11-00055-f002:**
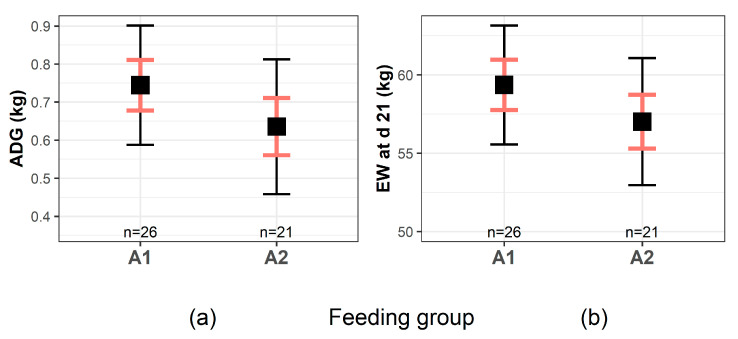
Least squares means for average daily weight gain (ADG, **a**) and end weight (EW, **b**) of calves receiving either A1- or A2-milk. Red and black bars indicate standard errors and upper-/lower levels of 95% confidence intervals, respectively. *n* is the number of calves per feeding group.

**Figure 3 animals-11-00055-f003:**
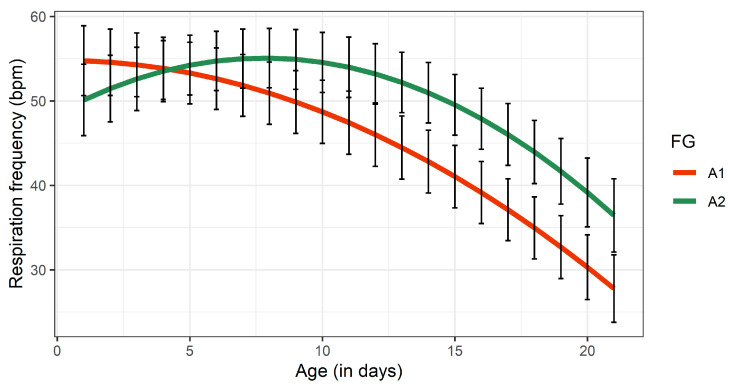
Least squares means for respiratory frequency of calves receiving either A1- or A2 milk during the experimental period of 21 days. Black bars indicate standard errors. FG, feeding group.

**Figure 4 animals-11-00055-f004:**
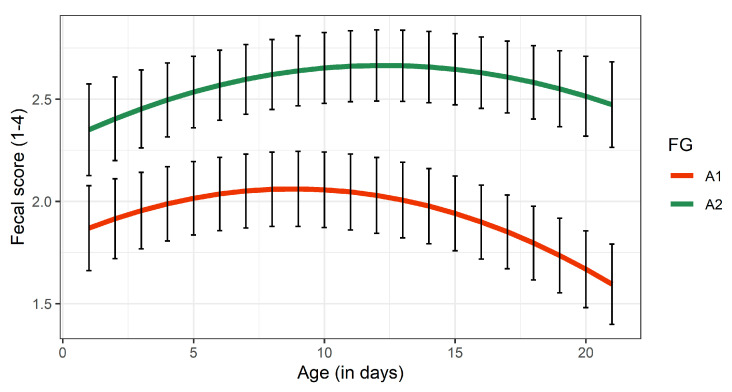
Least squares means for fecal scores of calves receiving either A1- or A2 milk during the experimental period of 21 days. Black bars indicate standard errors. FG, feeding group.

**Figure 5 animals-11-00055-f005:**
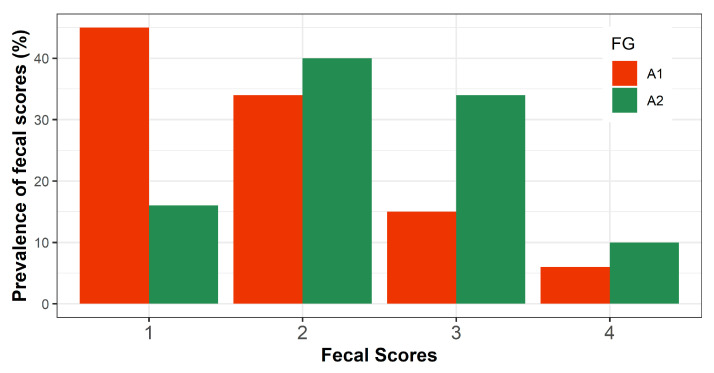
Prevalence of fecal scores of calves receiving either A1- or A2-milk.

**Figure 6 animals-11-00055-f006:**
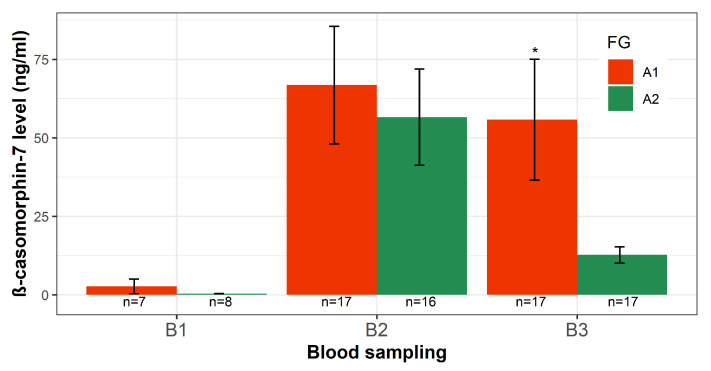
β-casomorphin-7 level in plasma of calves per feeding group (FG) at a different time of blood sampling. B1 = day 0; B2 = day 3; B3 = day 21. * *p* < 0.05, refers to *t*-test comparison between both feeding groups for each blood sampling. Black bars indicate standard errors.

**Table 1 animals-11-00055-t001:** Descriptive statistics for birth weight, average daily weight gain, body temperature, and respiratory frequency per breed group.

		BW (kg)		ADG (kg/day)		EW (kg)		dMI (L)		BT (°C)			RF (bpm)
	*n*	Mean	SD	Mean	SD	Mean	SD	Mean	SD	Mean	SD	Mean	SD
GH	9	44.30	3.05	0.71	0.22	59.72	4.77	6.78	0.79	38.8	0.51	51.6	23.20
GS	33	43.70	6.51	0.65	0.16	57.81	7.70	6.53	1.09	38.9	0.41	44.4	18.00
GH × GS	5	41.11	6.37	0.65	0.19	55.21	3.77	6.64	1.10	38.8	0.46	49.6	19.80

*n*, number of calves; GH, German Holstein; GS, German Simmental; SD, standard deviation; BW, birth weight; EW, end weight; ADG, average daily weight gain; dMI, daily milk intake; BT, body temperature; RF, respiratory frequency; bpm, breaths per minute.

**Table 2 animals-11-00055-t002:** Allele and genotype frequencies for β-casein by breed groups.

		Allele Frequencies			Genotype Frequencies				
	*n*	A1	A2	B	A1A1	A1A2	A2A2	A1B	A2B
GS	41	0.400	0.575	0.025	0.050	0.700	0.200	0.000	0.050
GH	8	0.500	0.500	0.000	0.125	0.750	0.125	0.000	0.000
GH × GS	62	0.492	0.475	0.033	0.164	0.590	0.180	0.066	0.000
Total	111	0.465	0.516	0.019	0.114	0.680	0.168	0.022	0.016

*n*, number of cows per breed; GS, German Simmental; GH, German Holstein.

**Table 3 animals-11-00055-t003:** Least squares means with corresponding standard errors (± SE), and *t*-test-statistic for milk intake, growth, and health parameters of calves receiving either A1-milk or A2-milk.

		Feeding Group		*p*-Value	*t*-Test-Statistic
		A1-Milk	A2-Milk		
Growth	dMI (L)	7.28 ± 0.12 ^a^	6.96 ± 0.11 ^b^	0.02	2.38
	ADG (kg/day)	0.75 ± 0.07	0.64 ± 0.08	0.07	1.87
	EW (kg)	59.40 ± 1.61	57.0 ± 1.73	0.07	0.76
Health	BT (°C)	38.94 ± 0.06	38.96 ± 0.05	0.75	−0.32
	RF (bpm)	46.60 ± 3.64	53.10 ± 3.50	0.13	−1.56
	FS	1.97 ± 0.18 ^a^	2.56 ± 0.17 ^b^	<0.001	−3.86

Least squares means within a row with different superscripts are significantly different (*p* < 0.05). dMI, daily milk intake; ADG, average daily weight gain; EW, end weight; BT, body temperature; RF, respiratory frequency; FS, fecal score; bpm, breaths per minute.

## Data Availability

The data presented in this study will be made available on reasonable request from the corresponding author.

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
