# Peer review of "Comparative Effects of Milk Containing A1 versus A2 β-Casein on Health, Growth and β-Casomorphin-7 Level in Plasma of Neonatal Dairy Calves"

_animals, 2020, doi:10.3390/ani11010055_

Round 1

Reviewer 1 Report

The manuscript is an interesting study about the effects of milk containing either homozygote A1 or A2 β-CN on health 27 and growth parameters of dairy calves.

My suggestions for improving this manuscript in attached file.

Reviewer 2 Report

Line 22-23: please include a "conclusion" statement. What is the main message from your data?

Line 28: what was the sex distribution?

Line 33-34: was solid feed provided? please be clear

Line 34-37: were calves developing intestinal digestive issues treated in some fashion? please indicate here

Line 38: indicate the actual protein content in the two groups

Overall, it is felt the abstract could include more actual data.

Line 87-93: somewhere here you need to state the hypothesis

Line 98: what does this statement really mean??

Line 152: for any ELISA assay please include inter- and intra-assay CV

Overall, good discussion of the data.

Reviewer 3 Report

Dear Authors,
only some minor suggestions:

170 - seems that the acronym FG has not been defined before in the text
227 - the acronym bpm is commonly associated to heart rate more than to Respiratory Rate

299 - if P<0.1 is not a typo the difference  between 49 and 56 bpm is not worthy of mention 

327 - is understandable that B1 samples were only 15 instead of 34 (B2,B3) but it would be better to clarify the reason for this difference.

Best Regards
AM

Round 2

Reviewer 1 Report

The authors corrected the manuscript according to the recommendations.